# Structural Characterization of Mono and Dihydroxylated Umbelliferone Derivatives

**DOI:** 10.3390/molecules25153497

**Published:** 2020-07-31

**Authors:** Rubén Seoane-Rivero, Estibaliz Ruiz-Bilbao, Rodrigo Navarro, José Manuel Laza, José María Cuevas, Beñat Artetxe, Juan M. Gutiérrez-Zorrilla, José Luis Vilas-Vilela, Ángel Marcos-Fernandez

**Affiliations:** 1GAIKER Technology Centre, Basque Research and Technology Alliance (BRTA), Parque Tecnológico de Bizkaia, edificio 202, E-48170 Zamudio, Spain; cuevas@gaiker.es; 2Departamento de Química Inorgánica, University of the Basque Country UPV/EHU, Apartado 644, 48080 Bilbao, Spain; estibaliz.ruiz@ehu.eus (E.R.-B.); benat.artetxe@ehu.eus (B.A.); juanma.zorrilla@ehu.eus (J.M.G.-Z.); 3Instituto de Ciencia y Tecnología de Polímeros (CSIC), Juan de la Cierva 3, 28006 Madrid, Spain; amarcos@ictp.csic.es; 4Interdisciplinary Platform for “Sustainable Plastics towards a Circular Economy” (SUSPLAST-CSIC), 28006 Madrid, Spain; 5Departamento de Química Física, University of the Basque Country UPV/EHU, Apartado 644, 48080 Bilbao, Spain; josemanuel.laza@ehu.eus (J.M.L.); joseluis.vilas@ehu.eus (J.L.V.-V.)

**Keywords:** coumarin, hydroxyl-modified coumarin, photophysical, thermal and structural characterization

## Abstract

Coumarin derivatives are a class of compounds with a pronounced wide range of applications, especially in biological activities, in the medicine, pharmacology, cosmetics, coatings and food industry. Their potential applications are highly dependent on the nature of the substituents attached to their nucleus. These substituents modulate their photochemical and photophysical properties, as well as their interactions in their crystalline form, which largely determines the final field of application. Therefore, in this work a series of mono and dihydroxylated coumarin derivatives with different chemical substituents were synthesized and characterized by UV-Visible spectroscopy, thermal analysis (differential scanning calorimetry (DSC) and TGA), ^1^H NMR and X-Ray Diffraction to identify limitations and possibilities as a function of the molecular structure for expanding their applications in polymer science.

## 1. Introduction

Coumarins (chromen-2-ones) are a family of benzopyrones widely distributed in nature. Since 1902, when Ciamician and Silber found that coumarin had the ability to be photoreactive, the photo-cyclodimerization and photo-cleavage of this product has received a lot of attention by several investigation groups [1,2,3,4]. Their structure are a class of lactones based on a benzene ring fused to α-pyrone ring, as can be seen in Scheme 1A [5,6]. Coumarins represent an important family of naturally occurring and/or synthetic oxygen-containing heterocycles, bearing a typical benzopyrone framework. One of the most important characteristics of coumarin derivatives is that they can undergo reversible photo-responsible reactions; depending on the type of irradiated wavelength, these moieties can yield a cyclobutane through dimerization or they can cleavage, reforming the double bond C=C. Thus, when irradiated at >300 nm, a [2 + 2] cycloaddition reaction takes place, forming a cyclobutane ring; in contrast, irradiating at 254 nm a photo-scission reaction leads to the original coumarin structures (Scheme 1B) [7,8,9,10]. 

Umbelliferone is a benzopyrone (also known as 7-hydroxycoumarin, hydrangine, skimmetine, and beta-umbelliferone) and belongs to the Coumarin family which is commonly found in plants [11]. The word “Umbelliferone” was originated from the plant which belongs to the Umbelliferae family. It includes significant herbs such as celery, carrot, garden angelica, sanicle, parsley, cumin, alexanders, big leaf hydrangea, fennel, asafoetida, Justicia pectoralis and giant hogweed. The phenolic coumarins, which are derived from plants, have been supposed to play a vital role in our daily life, due to their antioxidant property and are taken in the human diet in the form of vegetables and fruit [12]. 

Under acidic conditions and low temperatures, highly activated phenols, such as resorcinol and β-carbonyl ester, easily yield the desired coumarins; this synthetic approach is based on the biosynthesis of umbelliferone (Scheme 2).

Moreover, coumarin and its derivatives are small molecular weight compounds that have demonstrated very interesting physical, chemical and pharmacological properties with broad applicability as biochemicals (drugs, cosmetics, dyes, antibacterials, etc.) [13]. The diverse oriented synthetic routes have led to very different derivatives with usefulness not only as biologically active agents [14,15,16,17] or optical materials [18,19,20,21], but in macromolecular chemistry they also can be seen as very diverse polymer backbones with photoreactive properties [7,22,23,24]. Owing to the strong demand imposed by various government organizations for the design of polymeric sustainable materials in a more circular economic model, nowadays coumarin derivatives have acquired relevant importance in this field.

To elicit photoactive polymers with self-healing properties, different coumarin derivatives have been previously incorporated in various polymeric backbones [25,26,27,28]. In the particular case of polyurethanes, some research groups have recently introduced different hydroxylated derivatives either within the hard segment (chain end or chain extender) or within the soft segment (coumarin functionalized polycaprolactone diols) [2,29,30,31,32,33,34,35]. More recently, we reported an outstanding three times increment increase in the tensile strength of polyurethanes with difunctional hydroxy-coumarins, which led to new irradiated polyurethanes having mechanical properties superior to any coumarin containing materials described in literature [35]. Motivated by this issue, other hydroxylated coumarins, with a high structural analogy, were also introduced into the polyurethane matrices [2,33]. Comparing the experimental data derived from those works, uneven behaviors were observed, for instance, some coumarin hydroxyl-derivatives could not be introduced into the soft segment or the photo-dimerization yields varied considerably, to name a few. 

Owing to the coumarin contents in the polyurethane formulations being too low, a systematic study has been focused on the isolated coumarins. Therefore, in the present work those isolated hydroxy-coumarin compounds with different functionalities and chemical structure were synthesized and exhaustively characterized to shed light and understanding on those differences. Despite the structural similarity and considering the variability of substituents in the C7 position of the coumarins, an understanding at the atomistic level of the electronic interactions that determine the spectroscopic properties of coumarins has also been carried out. Single-crystal X-ray diffraction analyses revealed the key role of weak supramolecular forces in the self-assembly of molecular species within the crystal packing. The oxygen-rich molecular structures allow O–H···O hydrogen bonds and C–H···O type contacts to be established, which is demonstrated in their capacity to undergo edge-to-edge self-association [36]. Moreover, the presence of two fused aromatic rings in the coumarin structure contributes, with π-π stacking interactions, to the robustness of the system. These structural studies were essential to relate their different arrangements with the thermal behavior observed in DSC analyses.

## 2. Results and Discussion

### 2.1. Synthetic Approach

The chemical structures of the monohydroxy and dihydroxy-derived coumarins have been collected in Figure 1. 

All studied coumarins studied are based on the umbelliferone core. Firstly, the HMC product was prepared and later different chemical reactions (etherification or esterification) were carried out on this product to achieve the rest of presented coumarins. These reactions were focused on the lateral group located at C7 carbon, which was varied to assess its impact on the performance and features of these photo-reactive systems. In essence, two types of coumarins have been proposed, monohydroxy and dihydroxy, these hydroxyl-functional groups could, in turn, be directly bound to the umbelliferone ring or separated by a spacer. The synthetic routes used for the preparation of these coumarin derivatives have been included in the Appendix A. Likewise, a complete spectroscopic characterization of each coumarin has also been included in the Appendix A. Additionally, it is important to note that the synthetic procedures were simple, easily scalable and yields were good in most cases. Furthermore, due to the absence of post-purification processes, this set of features are key elements for the application of these coumarins in Industry. Indeed, the quality of the obtained products following these synthetic routes were high enough in order to incorporate them directly into the polyurethane formulations, leading to polymer coatings with interesting performances [33]. 

On the other hand, despite the high purity of these products during their synthesis, additional recrystallization steps were required for structural analysis by X-ray diffraction. Crystals suitable were obtained by dissolving the final products in their corresponding hot solvents (ca. 90 °C) and leaving the resulting solutions to slowly evaporate in an open container. First attempts were carried out by using the solvents of the synthetic procedure, but some of the cases did not yield crystals of enough quality. Thus, mixtures of solvents with different polarities were employed for the recrystallization. The longer the aliphatic chain of the substituent, the lower the polarity of the molecule. Due to this fact, coumarins with shorter substituents (HMC and DHMC) were found to crystallize better in solvents with higher polarity than coumarins with longer substituents (HEOMC and DHEOMC). That is, single crystals of HMC were obtained in the most polar solvent mixture, EtOAc:EtOH (3:1), whereas those of DHEOMC were isolated from the most nonpolar solvent mixture, EtOAc:diethyl ether (1:1). The remaining coumarins were easily recrystallized from EtOAc.

### 2.2. UV Experiments

Owing to strong absorption of UV light shown by coumarins, in the first set of experiments, the UV spectra of prepared coumarins in aqueous solution were recorded for the first time. 

In Figure 2, the absorption UV-spectra of pristine coumarin compounds (without irradiation) are shown. The concentration of these solutions ranged from 0.2 to 0.4 mM. In all cases, absorption of coumarins showed a π-π* transition between 260 and 300 nm attributed to electrons of the conjugated benzene nucleus and another π-π* transition between 310–340 nm assigned to pyrone nucleus [30]. Only in the case of DHMC, were these two transitions completely distinguishable, for the rest of coumarins the transition of the benzene ring appeared as a shoulder on the heterocycle transition band. However, the pyrone-associated transition is much more severely affected by UV radiation, while the transition of the benzene ring is practically unaltered. 

It is well known that UV-radiation markedly affects the reversible dimerization process of coumarin (Scheme 1). Thus, depending on the wavelength used as the source of excitation, this equilibrium shifts to one side or the other. When the aqueous solutions were irradiated with a set of five lamps of 354 nm, photo-dimerization reaction was performed, however, with a set of five lamps of 254 nm, photo-cleavage reaction was induced. 

The absorption UV-Vis spectra of HEOMC at 354 nm (A) and 250 nm (B) are depicted in Figure 3. In the photo-dimerization reaction (Figure 3A), the maximum transition at 320 nm gradually decreased in intensity with the irradiation time. As this band was associated with the heterocycle ring, the double bond of the pyrone core progressively disappeared, leading to the formation of a cyclobutene ring by cycloaddition [2 + 2]. In contrast, during photo-cleavage (Figure 3B), the double bond was restored and the absorption peak with the maximum recovered. Additionally, as the intensity of the band at 320 nm gradually decreased, the transition of the benzene ring was easy to detect, because it remained invariant with UV radiation.

The photoreactivity feature of characterized solutions can be quantitatively described by the time dependence of the maximum peak height [30]. Photo-dimerization degree was estimated from Equation (1), where A_t_ shows the absorbance at maximum peak at time t, and A_o_ the original absorbance at maximum wavelength prior to 354 nm exposure. When aqueous solutions were exposed to 254 nm, in order to characterize the recovery percentage, it used Equation (2); where A_∞_ denotes the absorbance after the solution exposed to 254 nm, A_∞_ shows the minimum absorbance at maximum peak after exposure to 350 nm UV light, and A_o_ has the same meaning as that in Equation (1).
(1)$$\%\;\mathrm{dimerization}\;\mathrm{degree}=\left(1\;-\;\frac{A_{t}}{A_{0}}\right)\;\times \;100\;$$
(2)$$\;\%\mathrm{photocleavage}\;\mathrm{degree}=\left(\frac{A_{\infty }^{\prime }{\;-\;A}_{t}}{A_{0}{\;-\;A}_{\infty }}\right)\;\times \;100$$

Significant differences in photo-dimerization and photo-cleavage were found in the first cycle of irradiation between studied coumarins. In Figure 4, the variation of the UV-absorbance with irradiation time of the four coumarin derivatives is shown. 

With respect to photo-dimerization, the DHMC coumarin product denoted a low dimerization degree: 13%. This could be based on the UV-spectrum of this derivative. As shown in Figure 2, the UV-spectrum of DHMC presented its absorption bands to higher wavelengths compared with its counterparts. Therefore, with an irradiation at 354 nm, the absorption of DHMC would be very weak and its dimerization would be hampered. Hence, to increase its dimerization degree, it would be necessary to irradiate at 313 nm (closer to the maximum peak of DHMC), but Seoane et al. demonstrated that irradiation at 313 nm gave a very strong irreversibility respect to photo-cleavage [2]. Optimum photoreversibility was achieved when irradiation was carried out with 354 and 254 nm sets of lamps. 

Regarding the other molecules studied (HMC, HEOMC, DHEOMC), the irradiation at 354 nm led to photo-dimerization yields of about 70%. Subsequently, only the HEOMC and DHEOMC derivatives largely recovered the initial absorbance by irradiation at 254 nm. In contrast, the HMC dimer was not able to significantly cleave in aqueous solution, and the absorbance values of this solution practically did not vary with irradiation at 254 nm. 

It is important to note that the HMC molecule has its functional group linked to the aromatic ring of coumarin, and this could be one reason to argue this behavior. With regard to how to improve the photoreactivity, there are some studies that denote that the addition of substituents to the coumarin dimer can improve the cleavage reaction efficiency, but Jiang et al. explained that it is not clear how the substituents modify the cleavage dynamics, or why they generally lead to enhanced efficiencies compared to the unsubstituted coumarin dimer [6].

One of the most important properties of coumarin derivatives is the photo-reversibility. This feature has been able to be studied in HEOMC and DHEOMC derivatives. In Table 1, the photo-dimerization and photo-scission yields for each cycle have been collected. Monohydroxy coumarin HEOMC progressively lost its photo-reversible capacity after the end of each cycle; in fact, after the third dimerization cycle (354 nm), only 30% of its coumarins had dimerized. In contrast, the dimers of dihydroxylated coumarin (DHEOMC) were easily cleaved after being irradiated with light at 254 nm. Although, its dimerization capacity was also depressed after each dimerization cycle, but the decrease was less pronounced than its monohydroxylated counterpart.

The kinetic of photo-dimerization (irradiation at 354 nm) and photo-cleavage (irradiation at 254 nm) for the HEOMC derivative is depicted in Figure 5. At the end of the photo-cleavage cycle, the absorbance value was lower than the starting point, so that the coumarin photo-reversibility was not perfect. The same effect was shown after the next cycle of photo-cleavage (end of cycle 2), where the absorbance was still even lower. Our findings suggested that photo-dimerization and photo-cleavage lost efficiency cycle by cycle. Indeed, the dimerization degree was slightly reduced whilst the cleavage dropped substantially as the number of cycles were repeated. Some authors attributed that the decrease in the photo-reversibility of coumarins could be due to the existence of an equilibrium between coumarin and its dimers, as well as the formation of non-cleavable dimers, because the lactone ring had probably opened [37].

### 2.3. Theory: Absorption UV-Vis Spectra

As studied coumarins have exhibited disparate UV-vis behavior and been previously used as photochemical crosslinking agents to obtain high-performance coatings, the need has been raised to understand the impact of the substituent on the electron densities of coumarins. 

Considering that the experimental UV-vis spectra were acquired in aqueous solution, the theoretical model should also include this effect. For this, the polarizable continuum model (PCM) was used, because this approach reproduced the experimental data with high precision at a low computational cost. Hence, we started with the optimized geometries of the ground state in the gas phase, which were then used as starting configurations for a geometric optimization of the molecules in aqueous solution using the PCM. Finally, these optimized settings served to determine the absorption properties of coumarins.

For all electrostatic potential surfaces (Appendix A), the carbonyl group (C=O) of the pyrone is the region with the highest concentration of electrons, making it the area with the lowest potential (in red). Furthermore, the coumarin family bearing two hydroxyl groups (DHMC, DHEOMC) presented a second region of low potential. This region was located in the other ester group and remained almost perpendicular to the coumarin ring. On the other hand, in all the coumarins studied, the regions with high potential correspond to hydroxyl groups, so that they will be the reactive centers during, for example, polymerization reactions.

In general, the maximum absorption for a molecule usually approximates the energy difference between the frontier orbitals (HOMO and LUMO). Eventually, the relative order of the calculated values of the HOMO–LUMO gap follows the same order with respect to the measured absorption peaks (Figure 6). Therefore, this observation could indicate that the dominant transition in UV-Vis spectra corresponds to the HOMO–LUMO transition for the C7-substituted coumarin family. However, in order to improve the description of the maximum UV-Vis absorption peaks, Time-dependent density functional theory (TD-DFT) calculations were performed for all coumarins, keeping the coumarin configuration frozen in the ground state according to the PCM. In this case, the theoretical results within the TD-DFT framework follow the same trend described above, but the maximum absorption peak is closer to the experimental one.

Additionally, a deeper analysis of the TD-DFT transitions yields very interesting information (Appendix A). For example, for all the cases studied, the first excited singlet state has the highest oscillator strength, except for coumarin DHMC, where the second excited singlet state also exhibits a significant contribution from the oscillator strength. As described above, the transition to the first singlet excited state corresponds mainly to the HOMO–LUMO transition. This transition has the same character for all C7-substituted coumarins except for DHMC, which also presents lower contributions than other transitions, such as HOMO-1 -> LUMO (2 -> 1) and HOMO-1 -> LUMO+1. (2 -> 2’).

On the other hand, the second singlet excitation state (S2) presents a very similar nature between the coumarins HMC, HEOMC and DHEOMC. For these cases, the transition is mainly governed by the HOMO-1 -> LUMO (2 -> 1’) transition and to a lesser extent by the HOMO -> LUMO +1 (1 -> 2’) transition. However, for DHMC, additionally in this transition to S2 there is also a slight contribution from the HOMO–LUMO (1 -> 1’) transition. Therefore, DHMC coumarin has a UV-Vis absorption spectrum with more significant differences compared to its counterparts, and the shape of this spectrum determines its behavior against UV radiation.

Through the morphological analysis of the frontier orbitals, which participate in the absorption processes, it should be possible to understand the optical properties of UV light absorption. Figure 7 depicts the frontier orbitals for the four C7-substituted coumarins. The two main frontier orbitals (HOMO and LUMO) of the coumarins studied are mainly extended along the heterocyclic ring defining delocalized π-orbitals. Only in the case of DHMC, is delocalization extended to the ester group located at C7; as for the rest of coumarin counterparts, the extension of the π-orbital is reduced to the C7 oxygen atom. This extension may be due to the aromatic ester nature of DHMC, while for the other ester coumarin (DHEOMC) it has a two-carbon spacer that breaks this conjugation. For the other two main orbitals (HOMO-1 and LUMO+1), they are preferentially concentrated in the benzene ring of coumarins, but only in the specific case of DHMC does the transition between these orbitals have a significant contribution.

### 2.4. Crystal Structures

Crystallographic data for compounds HMC, HEOMC, DHMC and DHEOMC are compiled in Table 2.
$${\;}^{a}R\left(F\right)=\sum \left|\left|F_{0}-F_{c}\right|\right|/\sum |F_{0}|.\;{\;}^{b}wR(F^{2})={\left\{\sum \left[w{\left(F_{0}^{2}-F_{c}^{2}\right)}^{2}\right]/\sum \left[w{\left(F_{0}^{2}\right)}^{2}\right]\right\}}^{1/2\;}$$

Thermal vibrations of all non-hydrogen atoms were refined anisotropically. Crystals of the DHMC derivative were systematically of much poorer quality than their analogues. Measurements on several crystals were made from several different batches in quest of a better set of crystallographic data but without any success. Although completeness was as low as 91%, and the used restrain/parameter ratio was considerably high, the best structural model was obtained solving the structure in the chiral Pc space group. The two coumarin molecules that form the asymmetric unit of DHMC were found to be involved in crystallographic disorder. The C and O atoms were refined with free population factors, resulting in a chemical occupancy of ca. 0.8 for the main form and 0.2 for the minor one. In order to model the disorder, several restrictions were applied for the thermal ellipsoids of C and O atoms belonging to the minor phase (ISOR). Some C–C and C–O bond lengths were also restricted to 1.54(2) and 1.43(2), respectively (DFIX). In all cases, hydrogen atoms of the organic molecules were placed in calculated sites using standard SHELXL parameters, whereas those from hydration water molecules were located in Fourier maps and restrained to O–H bond lengths of 0.84(2)Å. For DHMC, H atoms belonging to hydroxyl groups were placed in the Fourier map and bond lengths and angles were restricted using DFIX and DANG commands.

Single-crystal X-ray diffraction experiments revealed that molecular structures of the synthetized compounds are in good agreement with those proposed from ^1^H-NMR studies. In all cases, the coumarin backbone consists of a completely planar benzolactone ring bearing a methyl group in the C4 position which displays different substituents at the C7 position (Appendix A). Our findings on the crystal structure of monohydrate HMC and the supramolecular interactions governing its crystal structure were aligned with previously reported works [38,39]. Using this system as a base, significant changes within supramolecular interactions in the crystal structure for the rest of studied coumarins were also observed, owing to the insertion of different substituents at position C7. Indeed, intermolecular forces involved in the crystal packing of HEOMC, DHMC and DHEOMC include π-π interactions established between aromatic ring, O–H···O hydrogen bonds and C–H···O-type contacts. Their geometrical parameters are compiled in Appendix A. All the bond lengths and angles are in concordance with those found in literature [36,40,41,42].

In a close analysis of the HEOMC crystal structure, we observed that HEOMC crystallized in the monoclinic space group *P*2_1_/*n* and its asymmetric unit contained one HEOMC moiety and one hydration water molecule. As shown in Figure 8, the coumarin molecules were packed antiparallelly forming columns along the crystallographic z axis through π-π stacking. These arrangements were involved in an extensive three-dimensional network of O_W_–H···O and O–H···O_W_ hydrogen bonds established between the hydroxyl group of this monohydroxy coumarin and the hydration water molecules. Additionally, weak C–H···O-type contacts linked adjacent columns along the [100] direction.

On the other hand, the crystal structure of DHMC belonged to the monoclinic Pc space group, including two coumarin moieties in the asymmetric unit. Both coumarins were involved in crystallographic disorder; therefore, the forms with higher occupancy were only represented in Figure 9. DHMC molecules were packed antiparallelly, forming columns along the crystallographic y axis and interacting through π-π stacking and O–H···O hydrogen bonds. These strong interactions involved the O atoms from the ester group and both hydroxyl moieties. Owing to these four O atoms within this dihydroxy-coumarin, one-dimensional arrangements were connected to each other by creating an extensive network of C–H···O contacts.

Finally, DHEOMC crystallized in the triclinic space group P-1; the lower symmetry in comparison to the other systems could come from the introduction of a flexible spacer between dihydroxyl groups and umbelliferone moiety. In the case of this dihydroxy-coumarin, the crystal structure showed a bidimensional character with ribbons that stacked along the [0–11] direction. These ribbons were constituted by double-chains of coumarins that interacted through O–H···O-type hydrogen bonds and involved O atoms from hydroxyl or ester groups of different DHEOMC moieties. Contiguous double-chains were linked to each other via π-π interactions along the crystallographic y-axis, and together with C–H···O contacts, completed the bidimensional arrangement (Figure 10).

Ultimately, despite the symmetry of the space group of the coumarins studied, it should be highlighted that one molecule of water was included within the crystalline structures of monohydroxylated coumarins, whilst their dihydroxylated counterparts showed anhydrous structures. Probably, the two hydroxyl residues and the ester functional group were able to establish interactions similar to those arranged by the water molecules, in the monohydroxylates, to stabilize the crystal structure.

### 2.5. Thermal Analysis

The crystal structure of coumarin derivatives was also studied by DSC. As shown in Figure 11A, initially, in the heating scan, DSC measurements for monohydroxy coumarins (HMC and HEOMC) showed a broad endothermic peak that was possibly due to the loss of the recrystallization solvents (ethyl acetate, ethanol) and hydration water that occurs at 25–80 °C and 60–110 °C, respectively. Nevertheless, the dihydroxy-coumarins (DHMC and DHEOMC) suffered this peak. This finding is in line with the crystallographic data discussed above.

Additionally, significant differences in the melting temperatures of each compound were found in the heating scan. Coumarins bearing one hydration molecule in their asymmetric unit were found to have higher melting points than their dihydroxy partners, being 188 °C for HMC and 150 °C for HEOMC. Indeed, anhydrous crystal structures (DHMC and DHEOMC) presented a similar melting point between them, but much lower than the other hydroxylated coumarin family (121 for DHMC and 123 °C for DHEOMC). Comparing these two dihydroxy coumarins, DHEOMC presented a slightly higher melting temperature, which could be due to the flexibility of the spacer between the hydroxyl groups and umbelliferone ring favoring intermolecular interactions slightly higher. 

As shown in Figure 11B, only monohydroxylated coumarins displayed an efficient crystal packing, and because of these compounds exhibited a crystallization peak during the cooling process. Moreover, this crystallization took place at lower temperatures than melting points (151 vs. 188 °C for HMC and 74 vs. 150 °C for HEOMC). This fact could be due to the kinetics of crystal nucleation and growth being very slow. 

On the other hand, the enthalpies found for the melting and recrystallization processes are collected in Appendix A. It can be observed that the HMC and HEOMC coumarins partially recrystallized in the cooling process as a result of the enthalpy values for this step (∆Hc) being lower than the melting process (∆Hm). Additionally, the recrystallized fraction of HMC was higher than HEOMC. This fact could be related, again, to the more efficient crystal packing shown by this monohydroxylated coumarin compared to its monohydroxy partner.

Apart from that, the study of the thermal stability of crystal structures was also completed by TGA. In Figure 12, all the TGA curves of the studied coumarins are shown. For monohydroxy coumarins (HMC and HEOMC) the thermal decomposition occurred in two main mass-loss steps, whilst for dihydroxy coumarins (DHMC and DHEOMC) the decomposition carried out in a unique mass-loss step. In HMC and HEOMC coumarins, the first decomposition step appeared around 60–100 °C, corresponding to the loss of recrystallization solvents and hydration water. This result concurs with the X-ray diffraction data and DSC measurements described above. 

As shown in Table 3, an experimental mass loss of 2.7 and 7.5 wt % were observed for HMC and HEOMC, respectively. In the case of HEOMC, this TGA analysis agreed with the theoretically calculated value of 7.6 wt % for the loss of one water molecule within its crystal structure. Nevertheless, the theoretically calculated value for the loss of one water molecule in HMC was 9.3 wt %, which did not agree with the experimental value (2.7 wt %). This fact may be due to a weaker interaction between HMC and water that provoked a water loss at room temperature. To corroborate this issue, in a detailed analysis of the DSC curves (Figure 11), an endothermic peak was observed at room temperature for HMC, whilst for HEOMC at up 60 °C.

After this first mass-loss, the stability of all coumarins remained unaltered until 250 °C. Hence, this high thermal stability results in suitable unalterability for bulk polymerization reactive extrusion at temperatures above 150 °C, which is an efficient and very common industrial method for manufacturing thermoplastic polyurethane with incorporated target photoreactivity [43]. At 250 °C, a second mass-loss was observed, related to the thermal degradation of the coumarin ring. Depending on the system, the temperature of this second process appeared to vary. In fact, onset degradation temperatures (T0) and the weight loss corresponding to each degradation step are collected in Table 3 for all molecules. It should be noted that the two dihydroxylated coumarins had the highest and lowest decomposition temperatures for the second stage. In this sense, the DHMC product showed the lowest decomposition temperature (246.0 °C), probably due to the aromatic nature of its ester group. Through a transesterification reaction, this aromatic ester is very susceptible to attack by any hydroxyl group. This fact has already been described previously. The thermal stability of DHEOMC increased up to 323 °C, mainly due to the aliphatic nature of its ester group. 

## 3. Materials and Methods

### 3.1. Characterization

Solution 1H NMR spectra were recorded at room temperature in a Varian Unity Plus 400 instrument using deuterated chloroform (CDCl_3_) or deuterated dimethylsulfoxide (DMSO-d^6^) as solvent. Spectra were referenced to the residual solvent protons at 7.26 or 2.50 ppm, respectively.

To understand the habit of the reversible photoreaction of the coumarin derivatives, irradiations were carried out in an ultraviolet crosslinker supplied by Ultra-Violet Products equipped with two sets of 5 lamps for 354 or 254 nm irradiations (354 and 254 nm are the wavelengths on the maxima of the irradiation spectra of the lamps, respectively). The samples (from 0.2 to 0.4 mM concentration) were prepared by dissolving each coumarin derivative in distilled water. Firstly, each solution was exposed to 350 nm light in order to produce a crosslinked version via photo-dimerization. Afterwards, all the lamps were changed to give out 254 nm light and photo-cleavage of the crosslinked solution took place. In order to characterize these reversible UV kinetics, UV experiments were performed in a Perkin Elmer Lambda 35 UV-Vis spectrometer. Absorbance of the coumarin solutions were measured from 385 to 210 nm.

Thermal properties of all samples (5–10 mg) were measured (in duplicate) by differential scanning calorimetry (DSC, 822e from Mettler Toledo) in aluminum pans under constant nitrogen flow (20 mL·min^−1^). Each sample was subjected to a heating/cooling cycle from 25 to 200 °C with a heating rate of 2 °C·min^−1^. Thermogravimetric analysis measurements were also performed twice using TGA/DSC1 from Metter Toledo, under nitrogen (50 mL min^−1^) from room temperature to 600 °C with a heating rate of 10 °C min^−1^, where approximately 20 mg of sample was required.

### 3.2. Single-Crystal X-Ray Diffraction

Intensity data were collected on an Agilent Technologies SuperNova diffractometer equipped with monochromated Mo Kα radiation (λ = 0.71073 Å) and an Eos Charge-couple device CCD detector in the case of HMC and HEOMC, and monochromated Cu Kα radiation (λ = 1.54184 Å) and an Atlas CCD detector for DHMC and DHEOMC. The collection was performed at 100(2) K for HMC and HEOMC and 150(2) K for DHMC and DHEOMC. Data frames were processed (unit cell determination, mathematical absorption correction, intensity data integration and correction for Lorentz and polarization effects) using the CrysAlis Pro software package [44]. The structures were solved using the OLEX2 program [45] and refined by full-matrix least-squares using SHELXL-2014/6 [46]. Final geometrical calculations were carried out with PLATON [47] as integrated in WinGX software package [48].

### 3.3. Computational Methods

Quantum chemical calculations were performed with Gaussian 09 software [49]. Initially, the ground states were obtained by geometrical optimizations with the hybrid functional B3LYP and 6-311+(2d,p) as the basis set chosen for all the atoms.

As the experimental UV-Vis spectra of coumarins have been obtained in aqueous solution, the solvent effects of water were considered with the conductor-like polarizable continuum model (PCM). The calculated absorption energy was defined as the energy difference between the ground state and the excited state at the optimized ground state geometry.

## 4. Conclusions

Coumarin derivatives are widely distributed within the plant families and/or synthetic analogs for different applications. In this work, two types of hydroxy-derivative coumarins have been extensively analyzed by different techniques. Firstly, it is important to note that the synthetic routes presented were simple, easily scalable and with high yields, which could arouse high interest in the industry.

DHMC coumarin showed an uneven behavior compared to its counterparts, showing two strong absorption bands by UV-Vis. This performance conditioned its photophysical behavior to the dimerization reaction at 365 nm. Through a detailed atomistic study, these two absorption bands were attributed to the transitions between the frontier orbitals (HOMO, LUMO) and the two closest (HOMO-1, LUMO+1). Nevertheless, for the rest of the coumarins, the main excitation strongly corresponds to a π-π transition between the frontier orbitals.

Single-crystal X-ray diffraction analysis also demonstrated how hydroxyl groups allowed weak supramolecular forces to be established within the crystal structure, which were key elements in describing the ability to experience edge-to-edge self-association. Additionally, despite the symmetry of the space group of each coumarin, monohydroxy-derivated coumarins (HMC and HEOMC) presented one water molecule within each of their crystal structures, while their dihydroxylated counterparts (DHMC and DHEOMC) showed anhydrous structures.

Moreover, significant differences in the melting temperatures of each compound were found in the heating scan, which were consistent with an efficient crystal packing described by X-ray analysis.

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
