# Peer review of "Structural Characterization of Mono and Dihydroxylated Umbelliferone Derivatives"

_molecules, 2020, doi:10.3390/molecules25153497_

Round 1

Reviewer 1 Report

Reviewer report for Seoane-Rivero et al. Structural characterization of monofunctional and difunctional coumarin derivatives

molecules-860212

This paper describes the synthesis, photophysical, crystallographic and thermal characterization of four coumarin derivatives. I understood that the authors intend to incorporate these derivatives as building blocks in polymers.

Although I have no doubts that the experimental work and results are sound from a technical point of view, I miss the originality and significance required to warrant publication. At the current stage, a limited set of coumarins has been synthesized through routine methods and characterized by routine techniques. I have serious trouble in recognizing the bigger picture of this research – what is the objective of this study? What is the perspective of incorporating these building blocks in polymers?

The paper requires a much better storyline in the introduction to convince a reviewer and a reader that the work described herein is indeed relevant. I would therefore recommend that the authors publish these results in connection with polymer synthesis and characterization results.

From a synthetic point of view, the compounds claimed as new require substantially improved characterization – just H-NMR is simply not enough, for new compounds this has to be supplemented by 13C-NMR, IR and HRMS. What the authors could do right now is to publish the crystal structures separately in Acta Cryst. Or a related dedicated journal.

I should also mention that the paper is very hard to read, because the structures of the four coumarins are not depicted in any scheme or graphic in the results and discussion section, nor are any explanations for the abbreviations provided.

I regret that I am unable to recommend publication of this article.

Reviewer 2 Report

In this work a series of four hydroxy coumarine derivatives (two of which had been synthesized previously) have been synthesized and characterized by different techniques, with the aim of correlating the physico-chemical properties and different arrangements with the thermal behavior.

Coumarine derivatives, due to their ability to undergo a reversible photo-induced dimerization reaction, are widely employed as additives in photoreactive polymers with potential light-induced self-healing properties.

Two of the coumarine derivatives are claimed to be new but they are not fully characterized (only 1H NMR is provided). At least m.p., 13C NMR and MS spectra of the new compounds should be provided.

This manuscript is very hard to read due to very bad English. I recommend full revision of the English language before publication, and major revision in the chemical characterization part (provide full chemical characterization of the new compounds). Some hints for the language revision are provided below, but these are not exhaustive. The full text needs to be revised by a Native English speaker.

Abstract, line 20: Remove “especially….medicine” and replace with “especially in”

Line 25: remove “innovative”, only two compounds (of the four) are new. The synthesis is not innovative as well.

Line 27: replace “RMN” with “NMR”

Page 1, line 41: replace “appearing” with “forming”. Photoscission? Better photocleavage

Line 42: replace “renders” with “leads to”

Page 2, lines 50-51: bad English, please rephrase

Line 52: replace “they” with “coumarine”

Line 54: replace “-“ with “a” (add a)

Line 59: remove dot before “)”

Line 58: add in brackets “also known as”

Line 64: replace “is taken” with “are taken”

Replace “in a human” with “in the human”

Lines 64-66: not clear, Scheme 2 is not properly described. Please rewrite.

Page 2, line 80: remove “and”

Page 3, line 84: remove “innovative”

Line 85: “Chemical composition???” What does it mean?

Page 3, lines 99-100: bad English, rephrase with “and yields were good in most cases”

Line 100: “this is important for application of these coumarine derivatives in Industry”

Line 111: make the acronym explicit (HMC, DHMC, HEOMC, DHEOMC)

Line 113: replace “EtAc” with “EtOAc”

Scheme 3 should be placed in the Introduction section

Lines 117-121 are repetitions of what is discussed in the Intro. This part has to be shortened.

Page 3 (section 2.1) put a scheme containing the structures of the synthesized coumarines.

Page 4, line 129: “in all cases”

Lines 130-131: Not clear. Please rephrase and refer to compounds using numbers.

Page 4, lines 145-146: bad English, sentence not clear.

Figure 2 is missing

Page 5 line 159: “In Figure 3, the scheme shows”….it is a Figure or a Scheme???

Page 5 lines 174-175: Bad English

Page 6, lines 188-189: Bad English

Page 7, lines 204-206: bad English

Line 209: Explain what “PCM” stands for

Line 211: Figure 5 should be in bracket

Line 286: Spell correctly Figure 8

Line 287: Structure of HMC was already reported, the same is stated in lines 291-292. Where is the innovation here?

Schemes 4-7: the syntheses are not new, please remove these schemes. It is not clear which is the innovation brought by the authors.

Conclusions line 549-550: bad english

Author Response

The responses are collected in the attached file

Reviewer 3 Report

Interesting paper on structural characterization of monofunctional and difunctional coumarin derivatives. There is some aspects that must be improved:

a) Why the title do not referred the synthesis of the referred coumarins;

b) The authors claimed in the title the structural characterisation of coumarins, but there is no C-13 NMR, MS and HRMS...

c) Scheme 2 it os not a synthetic scheme, but a biosynthetic scheme

d) Figure 2 is missing

e) There is no discussion about the synthetic work carried out

f) The authors must send a grea number of tables and figures for SI, since they are not important in the discussion.

After these changes the manuscript can be accepted for publication

Author Response

Please find our comments in the attached file

Round 2

Reviewer 1 Report

The authors have substantially improved their manuscript with regard to the quality of presentation; however, I still don't think that the work is overly significant and it would have been better placed in a broader context, together with the relevant polymer aspects. I'm not sure if the manuscript will attract substantial impact, normally I would have rated the ms somewhere between average and low; but I would give the authors the benefit of doubt and - although with some hesitation - recommend publication of the work as it is, because from a technical point of view the ms is solid.

Reviewer 2 Report

The required changes have been done by the authors, so I think that this paper can be published in Molecules in its revised version.

Reviewer 3 Report

The authors have improved the quality of the manuscript, taking into consideration the comments and criticisms of reviewers.

The manuscript must be accepted in the present form.